# Computational Modeling of Spinal Locomotor Circuitry in the Age of Molecular Genetics

**DOI:** 10.3390/ijms22136835

**Published:** 2021-06-25

**Authors:** Jessica Ausborn, Natalia A. Shevtsova, Simon M. Danner

**Affiliations:** Department of Neurobiology and Anatomy, College of Medicine, Drexel University, Philadelphia, PA 19129, USA; nas29@drexel.edu

**Keywords:** computational modeling, neuronal control of locomotion, spinal cord, central pattern generator, interneurons, sensory feedback

## Abstract

Neuronal circuits in the spinal cord are essential for the control of locomotion. They integrate supraspinal commands and afferent feedback signals to produce coordinated rhythmic muscle activations necessary for stable locomotion. For several decades, computational modeling has complemented experimental studies by providing a mechanistic rationale for experimental observations and by deriving experimentally testable predictions. This symbiotic relationship between experimental and computational approaches has resulted in numerous fundamental insights. With recent advances in molecular and genetic methods, it has become possible to manipulate specific constituent elements of the spinal circuitry and relate them to locomotor behavior. This has led to computational modeling studies investigating mechanisms at the level of genetically defined neuronal populations and their interactions. We review literature on the spinal locomotor circuitry from a computational perspective. By reviewing examples leading up to and in the age of molecular genetics, we demonstrate the importance of computational modeling and its interactions with experiments. Moving forward, neuromechanical models with neuronal circuitry modeled at the level of genetically defined neuronal populations will be required to further unravel the mechanisms by which neuronal interactions lead to locomotor behavior.

## 1. Introduction

Locomotion is one of the most fundamental forms of movement and a prerequisite for any animal to interact with its environment. Neuronal control of locomotion involves interactions within and between neuronal networks in the spinal cord, supraspinal control centers, the musculoskeletal system, and sensory feedback [1,2,3,4]. Command signals from supraspinal centers and sensory information from the periphery are integrated by the spinal circuitry, which creates the locomotor rhythm and pattern and controls interlimb coordination. 

While much of what we know about the spinal locomotor system has been learned by studying reduced preparations (e.g., in spinalized, decerebrated, deafferented, or immobilized animals and in isolated spinal cord preparations), the spinal locomotor circuitry has frequently been treated as a black box with only limited access to its components. With the advent of targeted genetic approaches, only recently has it become possible to further dissect this black box and relate distinct neuronal populations to core locomotor functions [1,3,5,6,7,8,9]. 

The inherent complexity of the spinal circuitry itself and its interactions with descending control and afferent feedback is difficult to capture with experimental methods alone. In addition, novel molecular and genetic interventions in awake, behaving animals only introduce further levels of complexity and tremendous amounts of data. 

We discuss how computational models have been vital to understanding mechanisms underlying experimental observations in the past and how the recent advances in molecular genetics have amplified their importance. We review literature on spinal locomotor circuitry from a computational perspective and focus on modeling studies that aim to uncover underlying mechanisms of spinal neuronal locomotor control at the level of neuronal populations and their interactions. We describe computational models based on experiments in reduced preparations and awake behaving animals—using traditional methods and molecular genetics. This is not an exhaustive review, but rather a collection of representative examples to highlight the importance and relevance of computational modeling before the development and in the age of molecular genetics (Figure 1). We conclude by outlining how new tools and experimental approaches will necessitate a co-evolution of computational models to harness the synergistic benefits of model–experiment interactions. 

## 2. Computational Models Based on Classical Experimental Studies

Neuronal control of locomotion lends itself to computational modeling, even without more detailed knowledge of the constituting elements and their properties since motor output—in the form of motor nerve or muscle recordings—can be readily measured and quantified. This led to early computational modeling efforts (e.g., [11,12,13,14,15,16,17,18]) to propose underlying architectures mostly based on observations of the motor output pattern and its naturally occurring variations, responses to surgical, pharmacological, and electrophysiological manipulations, as well as single-cell recordings of spinal interneurons and motoneurons [19,20].

Spinal circuits can produce rhythmic locomotor-like activity in the absence of afferent feedback and rhythmic descending drive. The existence of these so-called locomotor central pattern generators (CPGs) was shown in pioneering studies by Graham Brown on the decerebrated and deafferented spinal cat at the beginning of the 20th century [21,22] and has since been confirmed as ubiquitous among vertebrates and invertebrates [23,24,25,26]. The reduction of the preparation to the isolated spinal cord—which represents a simpler but functional unit—enabled the first analyses of network function [27,28,29,30,31] and inspired early conceptual models of spinal locomotor circuits. The first of these conceptual models was already proposed by Graham Brown in his pioneering studies [21,22] in which he suggested mutually inhibiting ‘half-centers’: one controlling flexor and the other controlling extensor muscles. To allow oscillations between the two half-centers, he proposed a hypothetical fatiguing mechanism in each center and reciprocal inhibition between them that would allow transitions between flexor and extensor activity. 

Graham Brown’s simple conceptual model and its further elaboration by Lundberg and colleagues have been immensely useful in providing a framework and guiding subsequent experimental research [32,33,34,35,36]. This benefit of guiding experimental studies is what we regard as one of the most important roles of models for advancing research. The model adequately described the alternating nature of flexor-extensor activation and could theoretically be extended to drive and coordinate other alternating activity patterns (e.g., left–right). However, subsequent experimental observations showed that the half-center model cannot account for the complex muscle activation patterns of limbed locomotion [4,37,38]. In addition, experiments suggest that the core rhythm-generating mechanism does not require reciprocal inhibition but that populations of ipsilaterally projecting glutamatergic interneurons are sufficient to generate rhythmic activity [6,39]. While the half-center model does not seem to be the major locomotor rhythm-generating mechanism in the mammalian spinal cord, it nonetheless had a profound influence on the field of central pattern generators across the animal kingdom, as circuits resembling a half-center organization have been shown to exist in many other rhythmic motor systems [40,41,42]. 

With the rise of computer models, theoretical and practical considerations of the dynamics of half-center oscillators have been the subject of a vast number of computational studies (e.g., [11,43,44,45,46,47,48,49,50,51,52,53,54,55,56,57,58,59,60,61]) and the biological underpinnings have been explored in many systems (e.g., [27,42,47,62,63,64,65,66]).

Following in Graham Brown’s footsteps and sparked by increasing experimental access to spinal locomotor circuits, several other conceptual models were put forward (for a summary see [67]). Some of these were studied using computational modeling at various levels of complexity. Two conceptual models—the unit burst generator model [4] and the two-level CPG model [68]—have been explored in a series of computational studies that have significantly influenced the field.

It is now clear that the flexibility of motor patterns cannot be explained by the original half-center model and this apparent complexity inspired the formulation of the unit burst generator model [4]. This model suggests that the spinal cord comprises sets of rhythm-generating circuits (unit burst generators) that are coupled via inhibitory and excitatory interneurons. Each unit burst generator controls a degree of freedom or a pair of antagonistic muscle groups actuating one joint. This includes articulation between spinal segments to allow axial movement while swimming and walking, the coordination between limbs to allow for the generation of different gaits, and the coordination of individual joints within each limb [3,4,20,69]. The proposed organization theoretically allows for the flexible coupling and uncoupling of individual degrees of freedom. This organization was inspired by experimental evidence that individual muscle groups may be functionally disconnected in a way that burst activity may stop in some muscles while bursting in others continues, phase relationships may change, or some muscles may display stable extra bursts [4,20,37]. 

The unit burst generator model has been extensively investigated for axial locomotion. Experiments and computational modeling of the lamprey spinal cord pioneered this line of research (reviewed in [20,70]). The coordination between axial segments during undulatory swimming movements of lampreys and many other species shows an interesting commonality, where the body forms approximately one full wave that travels from rostral to caudal segments during forward swimming [71,72,73,74,75]. To maintain this full-body wave as locomotor frequency changes, the phase differences between activities in subsequent segments must remain constant. The question of how this constant phase-lag can be generated was explored in an exemplary interplay between experimental and computational studies. Early computational work by Kopell and Ermentrout suggested possible underlying mechanisms and guided subsequent experiments [76]. Using abstracted chains of oscillators, the modeling study predicted that constant phase-lags between segments can be generated by two mechanisms: asymmetric coupling of adjacent segmental oscillators and gradients in inherent segmental frequencies (the frequency a segment would generate when isolated from the rest of the nervous system). Evidence for coupling asymmetries was demonstrated in several experimental studies (reviewed in [75]). At the same time, the feasibility of frequency gradients between spinal segmental oscillators was demonstrated in the isolated spinal cord of the lamprey [77,78].

Computational modeling studies based on experimental data in the lamprey then further explored how a distributed organization of unit burst generators along the spinal cord can endow the spinal networks with the flexibility to switch between coordination patterns (such as in forward and backward swimming) while simultaneously allowing the characteristic expression of constant phase differences between segments [20,79,80,81]. Intersegmental coordination has been studied extensively for multiple model systems through computational modeling in several contexts and at various levels of complexity (e.g., [15,80,81,82,83,84,85,86,87,88]).

While a unit burst generator organization was shown for the stick insect [70], evidence for the existence of individual unit burst generators for intralimb coordination in vertebrates is not clear [70]. (Grillner and Kozlov discuss the applicability of the unit burst generator model for intralimb coordination in mammals in this Special Issue [89].) However, modeling studies show that a unit burst generator organization could coordinate limb rhythms with axial motor patterns [69,90,91], and spontaneous deletions in turtle scratch movements have been interpreted using the unit burst generator framework [92,93]. There is also experimental evidence for independent oscillatory units evoking flexor and extensor nerve activities of each limb [69,94,95]. 

Another influential model in the field, the two-level CPG model, was motivated by experimental observations during ‘fictive locomotion’ in immobilized and decerebrated cats [68,96,97,98]. With their two-level CPG model, McCrea and Rybak proposed that the spinal CPG for each limb has a two-level organization where a rhythm generator circuit sets the frequency of the locomotor rhythm and then drives a pattern formation network that shapes and coordinates the firing pattern of the specific muscle groups that actuate limb joints. 

Two lines of evidence motivated the concept of a two-level architecture for the locomotor CPG. First, certain types of afferent stimulation alter flexor or extensor activity of the ongoing cycle during fictive locomotion in the cat but do not impact the overall phase of the walking cycle [99,100,101,102]. Second, both resetting and non-resetting deletions (missing motor bursts in a group of synergistic muscles) occur during fictive locomotion in the cat [103]. Resetting deletions are characterized by a permanent shift in the phase of the locomotor rhythm after the deletion. The more common non-resetting deletions show no effect on the overall locomotor rhythm, and the post-deletion activity resumes at the appropriate phase. The two-level CPG model reproduces these results and offers a mechanism where afferent inputs and descending drive can act either on the rhythm generator or pattern formation level, or both. In this framework appropriately channeled afferent feedback or descending drive could allow for the flexible control and coupling of different parts of the motor pattern. Several computational modeling studies have shown the utility of this model architecture for the generation of realistic motor patterns for interlimb and intralimb coordination [104,105,106,107,108,109] and many subsequent experimental studies have interpreted their data within the framework of a two-level architecture [1,6,8,39].

The two-level CPG model has also been continuously updated alongside new experimental evidence. For example, the model was modified to incorporate an asymmetric, flexor-driven organization of the locomotor rhythm generator, which was originally proposed by Pearson and Duysens ([110,111,112], swing generator model). This model suggested a dominant role of rhythmic flexor activity, such that the flexor center is intrinsically rhythmic, while the extensor center is in a tonic mode and only exhibits rhythmic activity due to inhibition from the flexor center [113,114,115]. This flexor-dominant organization allows the model to reproduce the asymmetric changes of the flexor and extensor phase durations with increased locomotor frequency observed in vitro [113,116] and in vivo [117,118]. 

While the flexor-dominant CPG model can explain several experimental observations [10,114,118,119,120,121], some experimental data do not seem to support it [103,122,123,124]. As is often the case, computational modeling studies are well suited to investigate these seemingly contradictory data and explore their possible relationships. For example, computational modeling shows that multiple operating regimes can exist in the same general network architecture assuming varying levels of external drive to flexor and extensor populations [45]. Thus, state-dependent modulation of the neuronal components of the rhythm generators could set different operating regimes appropriate for the locomotor task at hand. For example, Latash et al. [125] showed that a model implementing a transition from a flexor-dominant regime at low locomotor frequencies to a classical half-center regime at high frequencies can explain the left–right asymmetries of split-belt locomotion in spinal cats [126,127]. 

The two-level CPG model was later expanded and explored when new transgenic techniques started to parse out the role of individual transcription factor-defined neuronal populations [10,113,118,119,120,121]. The advances in computational modeling that have been enabled by these novel techniques are outlined in the next section of this review.

All the models discussed here have been influential in providing frameworks for experimental studies and are not necessarily mutually exclusive and might coexist. For example, a similar organization to that of the unit burst generator model could underly the control and coordination of axial movement, and the coordination between limb and axial movements [3,20,69,91,128]; at the same time, control of individual joints within each limb might be organized as described by the two-level CPG model. Both the two-level CPG model and the unit burst generator model retain the basic mutual inhibitory organization of the half-center oscillator model for the coordination of antagonistic muscle groups. Even within the two-level CPG model framework, some of the microcircuits controlling different synergies might be able to produce rhythmicity when isolated experimentally [128]. A thorough computational exploration of the overlap and differences of these models would be useful for guiding further experimental studies. 

## 3. Computational Models Based on Molecular and Genetic Data

Novel molecular genetic methods have led to tremendous advancements in the study of spinal locomotor circuits [1,6,7,129,130]. Specifically, progress in mouse genetics allows for the identification and manipulation of elements of the circuitry. In combination with electrophysiological and imaging methods, it is possible to dissect the spinal locomotor network and link genetically identified neuronal populations to physiological markers and even distinct behaviors [1]. These methods provide a possibility to directly probe the constituting elements of the spinal locomotor circuitry and thereby break open the black box.

These experimental advances led to a new generation of computational modeling studies. The ability to manipulate individual transcription factor-defined neuronal populations allows computational models to study mechanisms at the network level and directly relate model predictions to experiments. We present a few examples of computational studies and the experimental results that led to them.

The cardinal class of V0 commissural interneurons has been implicated in controlling alternation between the left and right limbs [1,116,117,131]. The left–right alternation of the locomotor activity exhibited by wild-type mice in vitro was replaced by left–right synchronization in mutants lacking V0 commissural interneurons [116]. Interestingly, mutants lacking only interneurons of the excitatory subtype V0_V_ exhibit in-phase left–right activity at high oscillation frequencies only, while left–right alternation is maintained at low frequencies [116]. In addition, mutants lacking inhibitory V0_D_ commissural interneurons do not exhibit alternating activity at lower locomotor frequencies and demonstrate alternation at higher frequencies [116]. Thus, V0_V_ commissural interneurons maintain left–right alternation at high frequencies, while alternation at low frequencies is maintained by interneurons of the inhibitory subtype V0_D_. This finding of a frequency-dependent role of V0 commissural interneurons in vitro is important because mice in vivo use left–right alternating gaits at low speeds (lateral-sequence walk and trot) and left–right (quasi-)synchronous gaits at high speeds (gallop and bound) [117,132]. Indeed, a similar effect of deletion of V0 commissural interneurons on left–right coordination was observed in vivo: mutants lacking all V0 neurons were only able to bound (left–right synchronization), while mutants lacking V0_V_ neurons lost the ability to trot but were able to walk at low speeds [117]. 

These experimental results clearly link genetically defined classes of commissural interneurons to distinct behaviors. Computer models were then essential to propose potential circuit organizations and mechanisms underlying the frequency-dependent control of left–right alternation by V0_D_ and V0_V_ commissural interneurons [113,115,118,119,133]. The circuits can be viewed as coupled oscillators—rhythm generators on each side of the lumbar cord coupled by commissural interneurons. This concept has proven useful in modeling left–right segmental or interlimb coordination in lamprey, salamander, and mammals [106,134,135,136,137,138,139]. When the commissural connections are relatively weak, the commissural pathways mainly affect the phase relationships between the left and right rhythm generators [140,141]. Different commissural pathways promote different phase relationships and their relative strengths determine left–right coordination [8,54,115]. By building on previous modeling studies [96,97,114,115], Shevtsova et al. [113] proposed a model with three parallel commissural pathways coupling the rhythm generators of the left and right sides. Two pathways mediated by V0_D_ and V0_V_ commissural interneurons promote left–right alternation. The model also predicts a pathway tentatively mediated by V3 commissural interneurons that promote left–right synchronization in the absence of V0 neurons. While the inhibitory V0_D_ and excitatory V3 commissural interneurons were suggested to directly couple the left and right rhythm generators, the excitatory V0_V_ interneurons were suggested to receive input from ipsilaterally projecting excitatory V2a interneurons and project to inhibitory neurons on the contralateral side of the spinal cord. The model predicts that the frequency dependence of left–right coordination by the parallel V0_D_ and V2a-V0_V_ pathways could be produced by differences in their frequency-dependent recruitment in the presence of neuromodulators [142]. The model [113] not only reproduces the in vitro data in mutants lacking commissural interneurons belonging to V0 or its subtypes [116] but is also consistent with several other experimental data—such as asymmetric changes of flexor and extensor phase durations with increasing frequency and the effect of V2a-interneuron removal [143] on left-right coordination [113,116]. The underlying mechanisms and dynamics were analyzed in more abstract models [133]. The model [113] provides the basis for further computational models of neuronal control of interlimb coordination in vivo.

As in most quadrupeds [144,145], gaits in mice can be characterized by the coordination between the limbs [117,132,145]. Thus, to model the mechanisms underlying speed-dependent gait expression and its changes in mutants lacking V0_V_ or all V0 commissural interneurons [116,117], interactions between the circuits controlling each of the four limbs had to be considered [139,146]. To this end, the in vitro model by Shevtsova et al. [113] was extended to include four rhythm generators—one for each limb [118,119]. Left–right coordination was controlled through commissural interneurons coupling the rhythm generators within the lumbar and cervical enlargements. Fore-hind coordination was at first modeled through tentative populations of long propriospinal neurons coupling the rhythm generators on each side of the cord [118]. The model was then extended with a set of genetically defined homolaterally and diagonally projecting long propriospinal neurons [119], when more detailed experimental data became available [147]. Drive from the brainstem in the model was implemented to control speed by acting on the flexor centers of the rhythm generators and interlimb coordination by acting on commissural and long propriospinal neurons. With these suggestions, the computational models [118,119] were able to reproduce speed-dependent gait expression in intact mice (from walk to trot and then to gallop and bound), in mice lacking V0_V_ (selective loss of trot) as well as all V0 commissural interneurons (only bound expressed) [116,117,118], and in mice lacking descending (cervical-to-lumbar) long propriospinal neurons (transient episodes of disordered left-right coordination) [119,147]. The model [119] predicts a detailed connectome of commissural and long propriospinal interneurons mediating interlimb coordination. For example, in addition to left–right alternation, V0_V_ commissural interneurons promote the synchronization of diagonal rhythm generators through long propriospinal connections, supporting the hallmark features of a trot. Finally, the modulation of commissural and long propriospinal neurons in the model [119] causes gait changes independent of speed, suggesting that these neurons are prime targets for supraspinal inputs or somatosensory feedback to control gait—implicitly predicting the possibility of separate pathways controlling locomotor speed and interlimb coordination.

These models of central control of interlimb coordination and gait expression [118,119] have proven very useful in interpreting and modeling [120] novel experimental data on brainstem control of locomotion [148,149,150]. Using viral and genetic tools, a series of studies identified key neuronal populations within the nuclei of the mesencephalic locomotor region (the cuneiform nucleus, CnF, and pedunculopontine nucleus, PPN) [148,149] and within the reticular formation (specifically the lateral paragigantocellular nucleus, LPGi) [150] that are involved in the control of speed and gait. Optogenetic stimulation of glutamatergic neurons in the PPN was shown to result only in slow locomotion (walk and trot), while their stimulation in the CnF resulted also in fast locomotion (including gallop and bound) [148]. Interestingly, inactivation of these glutamatergic neurons in the PPN during concomitant activation of the CnF caused the transition from trot to gallop and bound to take place at lower locomotor speeds [148]. Both neuronal populations in the CnF and PPN have projections to the LPGi, which in turn contains neurons with projections to the spinal cord. Graded stimulation of glutamatergic neurons in the LPGi also drove locomotor speed and gait across the full range, from walk to trot and then to gallop and bound [150]. 

These experimental results were compatible with a previous model [119] and could be explained by incorporating brainstem circuitry with the suggestion that there are two parallel descending pathways from the brainstem to the spinal cord [120]: one controlling locomotor speed and the other interlimb coordination. The former involves both the CnF and PPN and excites the flexor centers of the rhythm generators, while the latter only involves the CnF and affects long propriospinal and commissural interneurons. Both pathways were hypothesized to be relayed by neurons in the LPGi. 

Thus, by leveraging previous models, a mechanistic model of the brainstem-spinal connectome was derived that is able to suggest underlying mechanisms for novel experimental data [120]. This line of modeling studies is exemplary in the fact that models were successively extended as new experimental evidence arose: this extension of previous computational models allowed for the investigation of new data in the context of previous knowledge. 

However, new experimental findings sometimes also lead to revisions of model assumptions. An excellent example is the case of V3 commissural interneurons. A computational model [113] predicted that these neurons promote left–right synchronization in fictive locomotion in the absence of V0 commissural interneurons through mutual excitation between the left and right flexor rhythm generator centers. This proposed V3 connectivity was retained in subsequent models of in vivo locomotion to provide the left–right synchronization necessary for gallop and bound [118,119,120]. However, novel experimental data were incompatible with the modeling prediction that V3 commissural interneurons provide mutual excitation between the flexor centers [10]: optogenetic activation of V3 neurons during fictive locomotion slowed down the ongoing rhythm, while the same manipulation in the model sped up the rhythm. This discrepancy inspired a revision of the original V3 connectivity. In the new model [10] (Figure 1b), V3 commissural interneurons receive input from the ipsilateral extensor center and excite both the contralateral extensor center as well as inhibitory interneurons that inhibit the contralateral flexor center. This model reproduced the new experimental data and was consistent with previous data showing changes in left–right activity following ablation of V0 or its subtypes [10,113,116]. Furthermore, V3 commissural interneurons still promoted left–right synchronization [10], as initially suggested by Shevtsova et al. [113]. (It should be noted here that V3 is a heterogeneous population [7,151,152,153] and both suggested connectivities might co-exist.) While experimental validation of model predictions confirms assumptions and solidifies our knowledge, their falsification challenges current hypotheses and facilitates subsequent model refinement, ultimately leading to the generation of new knowledge. This underscores that the main value of computational modeling is not to generate an absolute truth but to derive testable and falsifiable predictions that then guide further experiments. We even argue that the falsification of modeling predictions could be more valuable than verification in moving the field forward. 

## 4. A case for Neuromechanical Models Based on Molecular and Genetic Data

Molecular genetic tools allow studying manipulations of neuronal components in awake, behaving animals [1,117,148,154,155,156,157,158]. Thus, in contrast to reduced preparations, the spinal circuitry is embodied and interacts with the musculoskeletal system and the environment through motor output and afferent feedback. To address these interactions in computational models, the neuronal locomotor circuits have to be connected to a model of the musculoskeletal system (Figure 1c). Such neuromechanical models have been instrumental in studying contributions of afferent feedback to the control of locomotion. For example, Ekeberg and Pearson [159] studied afferent feedback mechanisms underlying phase transitions; Yakovenko et al. [160] studied the contribution of proprioceptive reflexes to locomotion; Aoi et al. [106,107,108] simulated how resetting of CPGs can lead to stable locomotion; Geyer et al. [161,162] showed how reflex chains, modulated by a state-machine, can generate functional locomotion; Di Russo et al. [163] studied the effect of feedback modulation on gait characteristics. Most of these models did not study the locomotor circuitry at the level of neuronal populations and their interactions but at higher levels of abstractions. An excellent example incorporating details of the neuronal circuitry is the study by Markin et al. [105], which investigated interactions of afferent feedback with a two-limb, two-level CPG model in generating stable locomotion.

A new generation of neuromechanical models with neural circuits based on molecular genetic data is now needed (Figure 1c). Using neuromechanical models with network models at the level of genetically defined neuronal populations [164] will enable us to directly simulate molecular genetic manipulations and observe behavioral changes (e.g., gaits, kinematics, muscle activities). The need for such detailed neuromechanical models is further amplified by the development of experimental methods to manipulate sensory components of the locomotor system [157,165,166,167,168,169,170,171]. This includes methods to access muscle spindles [169,172]—even spindles of individual muscles [170]—, proprioceptive feedback [169,173], and interneurons mediating presynaptic inhibition [171] or other sensory processing [166,167]. The use of these types of integrated neuromechanical models will not just allow simulation of perturbations to any part of the system, but also enable the prediction of resulting temporal dynamics of neuronal activity, muscle activations, and limb kinematics (Figure 1c and Figure 2). 

## 5. Conclusions

Computational modeling has been an integral part of the study of locomotor circuitry for several decades and many fundamental insights have resulted from synergistic interactions between models and experiments. Recent advances in molecular genetics have moved the whole field forward. Since neuronal populations and their interactions are at the core of many computational models, tools to identify and manipulate distinct populations have moved experimental methods to the same level of detail as the computational models. Thus, this new level of circuit manipulation now allows to closely relate models and experimental results (Figure 1), which makes a tight integration of computational modeling and experiments even more important: computational models can provide proofs-of-concept for hypotheses, identify missing information, and derive predictions; experiments can then elucidate the identified gaps in our knowledge and test modeling predictions; the experimental results then lead to model (and hypothesis) refinement; and the process can be repeated iteratively (Figure 2). We believe that such collaborative, integrated research programs will be essential to further disentangle the organization and function of the embodied spinal locomotor circuitry.

## Figures and Tables

**Figure 1 ijms-22-06835-f001:**
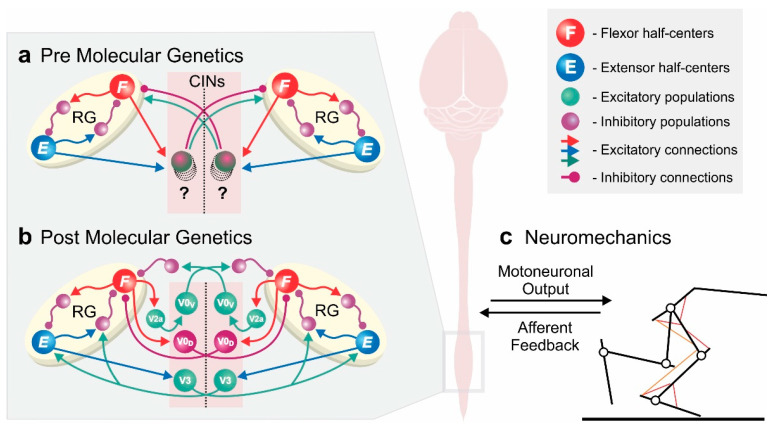
Conceptual overview of computational models of spinal locomotor circuitry. (**a**,**b**) Model schematics of two rhythm generators coupled by commissural interneurons (CINs), controlling and coordinating left and right rhythmic activities—based on classical experimental studies (**a**) and incorporating molecular genetic data (**b**). Experimental data of genetically modified animals allowed for models to disentangle the network of commissural interneurons coupling the rhythm generators on each side of the spinal cord. (**c**) Integrated models of neuronal locomotor circuits with the musculoskeletal system are needed to simulate molecular genetic manipulations in in vivo experiments and mechanistically relate them to behavioral changes. Schematic in (**b**) has been adapted from Danner et al. [10].

**Figure 2 ijms-22-06835-f002:**
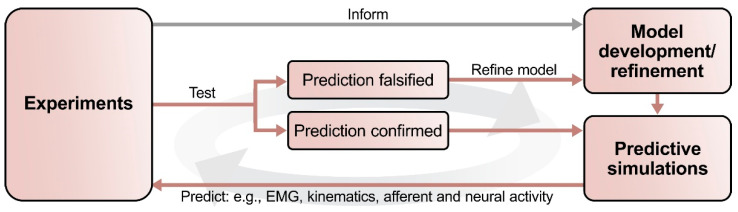
Outline of the suggested process of interactions between computational modeling and experimental studies. First, a set of experimental data informs the development of the initial computational model. This model should reproduce the key findings of the experimental studies. Predictions can then be generated by simulating experimentally testable conditions and manipulations that were not part of the initial set of experimental data. Next, these modeling predictions can be tested experimentally: experimental falsification of a modeling prediction will provide valuable information to refine the model; their confirmation will improve the validity of the model. When applied iteratively, this process provides an explicit and consistent theoretical framework for experimentation, thereby reducing the number of necessary experiments while simultaneously increasing the information gained per experiment.

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
