# Peer review of "Computational Modeling of Spinal Locomotor Circuitry in the Age of Molecular Genetics"

_ijms, 2021, doi:10.3390/ijms22136835_

Round 1
Reviewer 1 Report
- The number of references in the manuscript should be limited; only the most recent and most relevant references should be quoted (line number 39, here 13 references are cited (1, 4-16), there are many others, ref. 17-24, 29-40, etc)
- Rewrite the sentence "Using examples....with experiments." it looks like the authors have performed the experiments/analysis.
- In a review article, tables and a visual or graphical representation are more appealing than the text component. To strengthen the paper's readability, the authors could add a few figures and tables.
Author Response
We thank the reviewer for the helpful comments. We addressed all comments and respond below to each of them separately.
R1: 1. The number of references in the manuscript should be limited; only the most recent and most relevant references should be quoted (line number 39, here 13 references are cited (1, 4-16), there are many others, ref. 17-24, 29-40, etc)
Response: We agree with the reviewer and have removed several references. The total number of references was reduced by 26.
R1: 2. Rewrite the sentence "Using examples....with experiments." it looks like the authors have performed the experiments/analysis.
Response: The sentence has been rephrased.
R1: 3. In a review article, tables and a visual or graphical representation are more appealing than the text component. To strengthen the paper's readability, the authors could add a few figures and tables.
Response: We added a figure illustrating the main concepts of this review paper. The figure illustrates the difference in the level of detail between models based on classical experimental data and those based on molecular genetics data. It highlights how the ability to genetically identify and manipulate distinct neuronal populations leads to a more detailed understanding of the circuit organization and an evolution of the models. Further, the figure also illustrates the concept of neuromechanical models. Such models are needed to study mechanisms of sensorimotor integration and relate neuronal activity to behavior. We also added a figure illustrating an ideal process of interactions between computational modeling with experiments (Figure 2).
Reviewer 2 Report
Overall, I consider that the review paper 'Computational Modeling of Spinal Locomotor Circuitry in the Age of Molecular Genetics' has the interest to be published in IJMS in its present form."Computational Modeling of Spinal Locomotor Circuitry in the Age of Molecular Genetics" is an interesting review that comprehensively covers the literature on simulation of the spinal cord neuronal circuits with special emphasis on the different cell subpopulations, characterized by molecular genetics, which is its most novel feature. Interesting for those who work halfway between cybernetics, neuroscience and genetics.
Author Response
We thank the reviewer for their time and efforts and are grateful for the enthusiastic endorsement.
Reviewer 3 Report
We thank the reviewers for their article: Computational Modeling of Spinal Locomotor Circuitry in the Age of Molecular Genetics, this is an important topic that is worth to highlight and it is timely. New advances in computational modeling have addressed many of the unknown fundamental key points in locomotor control. This is mainly due to the fact that the integration of experimental studies and computational modeling has become possible, allowing the exploration of the rationale behind spinal locomotor circuitry. The title of this review ‘Computational Modeling of Spinal Locomotor Circuitry in the Age of Molecular Genetics’ is direct, clearly introducing the phenomena of interest and the key purpose of the paper. As for the abstract, it is concise, providing a clear emphasis on how stable locomotion is achieved through integrated supraspinal commands and afferent feedback signals making it easy to grasp, especially for those who are unfamiliar with the topic. The main purpose of the review being the spinal locomotion circuitry from a computational perspective is clearly stated, giving the reader a sense of what is going to be reviewed in the paper.
There are several points of strengths
Every statement & evidence introduced in the paper is properly cited
Information is presented in an organized manner (sections with titles) allowing the reader to easily tackle the huge amount of information presented in the review
The conclusion is well written, reinforcing the main objective of the paper and the key points in a summarized manner
Below are specific comments that need to be addressed
- Advancements in computational approach is beneficial, but the authors seem biased towards the topic. Lack of contrary data? Limitations? Strengths & weaknesses?
- The authors did not add any figures to the paper (some information would be better depicted if some sort of figure or diagram was included). For instance, since the paper is about computational modeling in the age of molecular genetics, a graphical timeline would better describe the various models that has been developed over time, and also depict any modification that has been made.
- Also, how have computational modelling advanced genetic studies in locomotor research? Since it is supposed to lead to discoveries that could aid in the development of more hypothesis. Can a figure be made for this also?
- Section 2.0, third paragraph (However, over time, experimental observations that characterized the 86 motor pattern ….across the animal kingdom)à Run-on sentences? Unclear sentence structure?
- Line 145-146: “multiple-different†is the same in this context, you can choose one
- Line 57 -216: Several of the computational models discussed within this line were not well detailed. For instance, the â€half-Center oscillator model€
- Lack of proper punctuation especially commas and spacing
Author Response
We thank the reviewer for the helpful comments. We addressed all comments and respond below to each of them separately.
R3: Below are specific comments that need to be addressed
Advancements in computational approach is beneficial, but the authors seem biased towards the topic. Lack of contrary data? Limitations? Strengths & weaknesses?
Response: In this paper, we review literature on computational models of the spinal locomotor circuitry. We highlight the role that modeling studies played and how they contributed to the current state of knowledge. Finally, we argue that computational models will be important and useful to move the field forward. We agree with the reviewer that any kind of computational model has weaknesses and limitations. But we do not argue that a specific model or type of model should be used. We argue for computational modeling as a general tool. Thus, it doesn’t make sense to describe weaknesses and limitations of computational modeling for our review because these depend on the type of model employed, on the equations used, and on how the models are interpreted. On the other hand, we clearly state that current models are insufficient to capture experimental data collected in awake and behaving mice; and we propose that neuromechanical models (neural network models integrated with a model of the musculoskeletal system) will be necessary to resolve this weakness of current models. We have carefully read and edited the text to ensure that these statements are clear.
R3: The authors did not add any figures to the paper (some information would be better depicted if some sort of figure or diagram was included). For instance, since the paper is about computational modeling in the age of molecular genetics, a graphical timeline would better describe the various models that has been developed over time, and also depict any modification that has been made.
Response: We added two new figures. Figure 1 illustrates the main concepts of this review paper. The figure illustrates the difference between models based on classical experimental data and those based on molecular genetics data. It highlights how the ability to genetically identify and manipulate distinct neuronal populations leads to a more detailed understanding of the circuit organization and an evolution of the models. Further, the figure also illustrates the concept of neuromechanical models: such models are needed to study mechanisms of sensorimotor integration and relate neuronal activity to behavior.
We also added a figure illustrating an ideal process of interactions between computational modeling with experiments (Fig 2; see below).
We have not added a figure illustrating all models discussed in the paper because there are too many models and some of them are very complicated. Such a series of figures would distract from the main messages of the paper. We believe that the added figures capture the main points made and add value to the manuscript.
R3: Also, how have computational modelling advanced genetic studies in locomotor research? Since it is supposed to lead to discoveries that could aid in the development of more hypothesis. Can a figure be made for this also?
Response: Mainly, general model concepts (half-center, two-level, or unit burst generator organization) have actively informed experimental studies as described in section 2. Close integration of experimental studies and computational modeling is rare. This is the main reason why we wrote this review; we wanted to advocate for such a process. To further clarify how computational modeling can impact and advance genetic studies, we added Figure 2 illustrating how we imagine the ideal process of interactions between computational modeling and experiments.
R3: Section 2.0, third paragraph (However, over time, experimental observations that characterized the 86 motor pattern ….across the animal kingdom)à Run-on sentences? Unclear sentence structure?
Response: We shorted and rephrased the sentence.
R3: Line 145-146: “multiple-different†is the same in this context, you can choose one
Response: Fixed.
R3: Line 57 -216: Several of the computational models discussed within this line were not well detailed. For instance, the â€half-Center oscillator model€
Response: We intentionally kept model descriptions brief. Our aim was to include enough detail to illustrate their usefulness in interactions with experiments without overwhelming the core message of the review. Detailed descriptions of the models have been undertaken in other reviews, which we took care to refer to. With this in mind, the level of detail in section 2 is comparable to that of the models based on genetic data (section 3). The difference is that models based on genetic data are much more complex and hence more text is required for their explanation.
For example, the description of the half-center model in the text is: “The first of these conceptual models was already proposed by Graham Brown in his pioneering studies [27,28], in which he suggested mutually inhibiting ‘half-centers’: one controlling flexor and the other controlling extensor muscles. To allow oscillations between the two half-centers, he proposed a hypothetical fatiguing mechanism in each center and reciprocal inhibition between them that would allow transitions between flexor and extensor activity.”
R3: Lack of proper punctuation especially commas and spacing
Response: We apologize for the errors in the original submission. We asked an English professor to proofread the manuscript and thoroughly edited the whole document to improve the language and the use of punctuation.